# DOMAIN ADAPTATION WITH FEATURE AND LABEL DISTRIBUTION CO-ALIGNMENT

## ABSTRACT

Unsupervised knowledge transfer has a great potential to improve the generalizability of deep models to novel domains. Yet the current literature assumes that the label distribution is domain-invariant and only aligns the feature distributions or *vice versa*. In this paper, we explore the task of *Generalized Domain Adaptation*: How to transfer knowledge across different domains in the presence of both shift in the feature and label distribution? We propose a feature and label distribution CO-ALignment (COAL) model to tackle this problem. Our model leverages prototype-based conditional alignment and label distribution estimation to alleviate the negative effect of the the types of domain shift respectively. We demonstrate experimentally that when both types of shift exist in the data, COAL leads to state-of-the-art performance on several cross-domain benchmarks.

## 1 INTRODUCTION

The success of deep learning models is highly dependent on the assumption that the training and testing data are *i.i.d* and sampled from the same distribution. In reality, they are typically collected from different but related domains, leading to a phenomenon known as *domain shift* (Quionero-Candela et al., 2009). To bridge the domain gap, Unsupervised Domain Adaptation (UDA) transfers the knowledge learned from a labeled source domain to an unlabeled target domain by statistical distribution alignment (Long et al., 2015; Tzeng et al., 2014a) or adversarial alignment (Tzeng et al., 2017; Ganin & Lempitsky, 2015a; Saito et al., 2018). Though recent UDA work has made great progress, it has mostly failed to address the case of Shift in Label Distribution, *i.e.*, a changing prior over the labels. Denote the input data as $x$ and output labels as $y$, and let the source and target domain be characterized by probability distributions $p$ and $q$, respectively. The majority of methods assume that the conditional label distribution is invariant ($p(y|x) = q(y|x)$), and only the *Shift in Feature Distribution (SFD)* ($p(x) \neq q(x)$) needs to be tackled, neglecting any potential *Shift in Label Distribution (SLD)* ($p(y) \neq q(y)$). However, recent theoretical work (Zhao et al., 2019a) has demonstrated that tackling SLD is crucial to solving the domain adaptation problem. SLD also occurs in real applications; for example, an autonomous driving system should be able to handle changing frequencies of pedestrians and cars when adapting from a rural to a downtown area.

In this paper, we propose **Generalized Domain Adaptation** (**GDA**), a more challenging but practical domain adaptation setting where the conditional Shift in Feature Distribution and Shift in Label Distribution are required to be minimized simultaneously. Specifically, in addition to the Covariate Shift assumption ($p(x) \neq q(x), p(y|x) = q(y|x)$), we further assume $p(x|y) \neq q(x|y)$ and $p(y) \neq q(y)$. The main challenges of GDA are: (**1**) SLD exists between the source and target domain, which hampers the effectiveness of mainstream domain adaptation methods that only minimize SFD, (**2**) aligning the conditional feature distributions ($p(x|y), q(x|y)$) is difficult in the presence of SLD, and (**3**) when the data in one or both of the domains are unequally distributed across different categories, it is difficult to train an unbiased classifier. An overview of GDA is shown in Figure 1.

Mainstream unsupervised domain adaptation aligns the marginal feature distributions of two domains by methods that include minimizing the Maximum Mean Discrepancy (Long et al., 2015; Tzeng et al., 2014a), aligning high-order moments (Zellinger et al., 2017; Peng et al., 2019a), or adversarial training (Tzeng et al., 2017; Ganin & Lempitsky, 2015a). These methods have achieved state-of-the-art performance on several domain adaptation benchmarks (Saenko et al., 2010; Venkateswara et al., 2017; Peng et al., 2017) which have significant SFD but limited SLD.

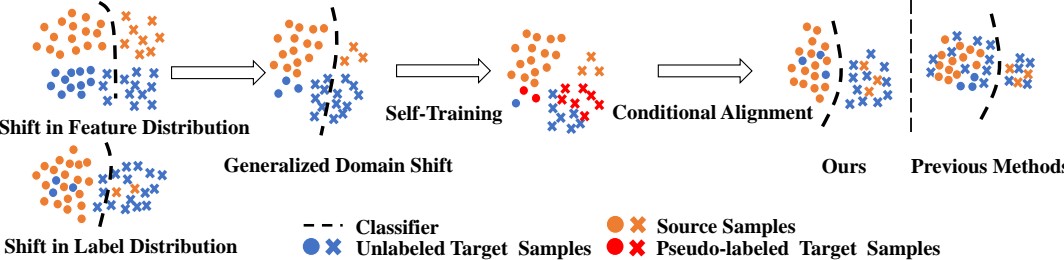

Figure 1: We propose the GDA setting, where we consider SFD and SLD simultaneously. To tackle this problem, we propose to use **self-training** to estimate and align the target label distribution, and use a prototype-based method for **conditional alignment**. In this way, we can better align the feature distribution of each category. Due to label shift, previous methods that learn marginal domain-invariant features will incorrectly align samples from different categories, leading to negative transfer.

These models are limited when applied to the GDA task as they only align the feature distribution, ignoring the real issue of SLD which exists in many real-world applications. Another line of works (Lipton et al., 2018a; Azizzadenesheli et al., 2019a) assume that only SLD exists ($p(y) \neq q(y)$) between two domains and the conditional feature distribution is invariant ($p(x|y) = q(x|y)$). These methods have achieved good performance when the data in both domains are sampled from the same data distributions but under different label distributions. However, these models cannot handle the GDA task as the label distribution is not well aligned. Taking steps towards minimizing the SFD and SLD simultaneously, several works (Zhang et al., 2013; Gong et al., 2016; Wu et al., 2019) provide a theoretical analysis with additional constraints on distributions $p$ and $q$, such as a linearity assumption between $p(x|y)$ and $q(x|y)$ (Zhang et al., 2013), which does not necessarily hold in many real applications. In addition, no practical algorithm which can solve real-world cross-domain problems has been proposed by these papers.

We postulate that it is essential to align the conditional feature distributions as well as the label distributions to tackle the GDA task. In this work, we address GDA with SFD and SLD **CO-ALignment** (**COAL**). Specifically, our approach diminishes the negative effect of SFD and SLD with *prototype-based conditional distribution alignment* and *label distribution estimation*, respectively. First, to reduce the SFD in the context of SLD, it is essential to align the conditional rather than marginal feature distributions, to avoid the negative transfer effects which are caused by matching marginal feature distributions, according to a theoretical proof from Zhao et al. (2019a) (illustrated in Figure 2). To this end, we propose a prototype-based method to align the conditional feature distributions of the two domains. The *source* prototypes are computed by learning a similarity-based classifier and the *target* prototypes are estimated by a minimax entropy algorithm (Saito et al., 2019b). Second, we align the label distributions in the context of SFD by estimating the target label distribution. We incorporate feature distribution and label distribution alignment into an end-to-end deep learning framework, as illustrated in Figure 1. Comprehensive experiments on standard cross-domain recognition benchmarks demonstrate that COAL achieves significant improvements over the state-of-the-art methods on the task of GDA.

The main contributions of this paper are highlighted as follows: (1) to the best of our knowledge, we provide the first practical solution and set of benchmarks for joint SFD and SLD in deep learning; (2) we develop an end-to-end CO-ALignment (COAL) framework which leverages conditional prototype-based alignment to solve the problem caused by SFD and label distribution estimation to solve the problem caused by the SLD; and (3) we deliver extensive experiments and analysis to show the effectiveness of COAL, and importantly, we empirically demonstrate that state-of-the-art methods fail to align conditional feature distribution in the presence of label distribution.

## 2 RELATED WORK

**Unsupervised Domain Adaptation** Domain adaptation aims to transfer the knowledge learned from one or more source domains to a target domain. Recently, many unsupervised domain adaptation methods have been proposed. These methods can be taxonomically divided into three categories (Wang & Deng, 2018). The first category is the discrepancy-based approach, which leverages different measures to align the marginal feature distributions between source and target domains.

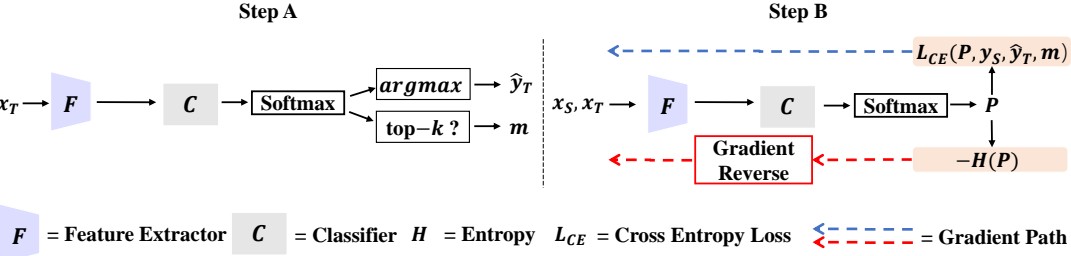

Figure 2: **Overview of the proposed COAL model**. Our model is trained iteratively between two steps. In step A, we forward the target samples through our model to generate the pseudo labels and mask. In step B, we train our models by *self-training* with the pseudo labels and mask to align the label distributions, and *prototype-based conditional alignment* with the minimax entropy.

Commonly used measures include Maximum Mean Discrepancy (MMD) (Long et al., 2017; Tzeng et al., 2014b), $\mathcal{H}$-divergence (Ben-David et al., 2010), Kullback-Lerbler (KL) divergence (Zhuang et al., 2015), and Wasserstein distance (Lee & Raginsky, 2017; Shen et al., 2017). The second category is the adversarial-based approach (Tzeng et al., 2017; Liu & Tuzel, 2016; Peng et al., 2019b) which uses a domain discriminator to encourage domain confusion via an adversarial objective. The third category is the reconstruction-based approach. Data are reconstructed in the new domain by an encoder-decoder (Bousmalis et al., 2016; Ghifary et al., 2016) or a GAN discriminator, such as dual-GAN (Yi et al., 2017), cycle-GAN (Zhu et al., 2017), disco-GAN (Kim et al., 2017), and CyCADA (Hoffman et al., 2018). However, these methods mainly consider aligning the marginal distributions to decrease feature shift, neglecting label shift. To the best of our knowledge, we are the first the propose an end-to-end deep model to tackle both feature and label shift between the source and target domains.

**Learning with Label Shift** Despite its wide applicability, learning under label shift remains under-explored. Existing works tackle this challenge by importance reweighting or target distribution estimation. Specifically, Zhang et al. (2013) exploit importance reweighting to enhance knowledge transfer under label shift. Recently, Lipton et al. (2018b) introduce a test distribution estimator to detect and correct for label shift. Based on it, Azizzadenesheli et al. (2019b) propose a regularized algorithm to better estimate the target label distribution. These methods assume that the source and target domains share the same generative distributions and only differ in the marginal label distribution. In this work, we explore transfer learning between domains under domain and label shifts. In this direction, Zhao et al. (2019b) introduce a theoretical analysis to show that only learning domain-invariant features is not sufficient to solve domain adaptation task since the label priors are not aligned. In related work, Wu et al. (2019) propose asymmetrically-relaxed distribution alignment to overcomes the limitations of standard domain adaptation algorithms which aims to extract domain-invariant representations. Panareda Busto & Gall (2017) propose *open set domain adaptation* where the categories in the training domain and testing domain are not fully overlapped. Cao et al. (2018b) introduce *partial domain adaptation* where the categories in the target domain are a subset of those in the source domain.

**Domain adaptation with self-training** Self-training methods utilize pseudo-labels to compensate for the lack of categorical information in the target domain. The intuition is to assign pseudo-labels to unlabeled samples based on the predictions of one or more classifiers. Long et al. (2013) jointly align the marginal and conditional distributions using an MMD loss with pseudo-label refinement. Saito et al. (2017) leverage an asymmetric tri-training strategy to assign pseudo-labels to the unlabeled target domain. Xie et al. (2018) propose to assign pseudo-labels to all target samples and use them to achieve semantic alignment across domains. Zhang et al. (2018) progressively enlarge the training data with pseudo-labeled target samples assigned by the classifier from the previous training epoch and retrain the model with the enlarged training set. Recently, Chen et al. (2019a) propose to progressively label the target samples and align the *prototypes* of source domain and target domain to achieve domain alignment. However, to the best of our knowledge, self-training has not been applied for DA with label shift.

# 3 CO-ALIGNMENT OF FEATURE AND LABEL DISTRIBUTION

In domain adaptation with shift in the feature and label distribution, we are given a *source* domain $\mathcal{D}_{\mathcal{S}} = \{(x_i^s, y_i^s)_{i=1}^{N_s}\}$ with $N_s$ labeled examples, and a *target* domain $\mathcal{D}_{\mathcal{T}} = \{(x_i^t)_{i=1}^{N_t}\}$ with $N_t$ unlabeled examples. We assume that $p(x|y) \neq q(x|y)$ and $p(y) \neq q(y)$. We aim to construct a end-to-end deep neural network which is able to transfer the knowledge learned from $\mathcal{D}_{\mathcal{S}}$ to $\mathcal{D}_{\mathcal{T}}$, and train a classifier $y = \theta(x)$ which can minimize task risk $\epsilon_T(\theta) = \Pr_{(x,y) \sim q}[\theta(x) \neq y]$.

Previous works either focus on aligning the marginal feature distributions (Long et al., 2015; Tzeng et al., 2017) or aligning the label distributions (Lipton et al., 2018b). These approaches are not able to fully tackle the domain adaptation with shift in the feature and label distribution as they only align one of the two marginal distributions. In this work, we tackle GDA with *prototype-based conditional alignment* and *label distribution estimation*.

## 3.1 PROTOTYPE-BASED CONDITIONAL ALIGNMENT OF FEATURE DISTRIBUTION

The mainstream idea in covariate-shift oriented methods is to learn domain-invariant features by aligning the marginal feature distributions, which was proved to be inferior in the presence of label shift (Zhao et al., 2019a). Instead, we propose to align the conditional feature distributions. To this end, we leverage a *similarity-based classifier* to estimate $p(x|y)$, and a minimax entropy algorithm to align it with $q(x|y)$. We achieve conditional feature distribution alignment by aligning the source and target prototypes in an adversarial process.

**Similarity-based Classifier** The architecture of our COAL model contains a feature extractor $F$ and a similarity-based classifier $C$. Similarity- or prototype-based classifiers perform well in few-shot learning settings (Chen et al., 2019b), which motivates us to adopt them since in label-shift settings some categories can have low frequencies. Specifically, $C$ is composed of a weight matrix $\mathbf{W} \in \mathbb{R}^{d \times c}$ and a temperature parameter $T$, where $d$ is the dimension of feature generated by $F$, and $c$ is the total number of classes. Denote $\mathbf{W}$ as $[\mathbf{w}_1, \mathbf{w}_2, ..., \mathbf{w}_c]$, this matrix can be seen as $c$ $d$-dimension vectors, one for each category. For each input feature $F(x)$, we compute its similarity with the $i_{th}$ weight vector as $s_i = \frac{F(x)\mathbf{w}_i}{T\|F(x)\|}$. Then, we compute the probability of the sample being labeled as class $i$ by $h_i(x) = \sigma(\frac{F(x)\mathbf{w}_i}{T\|F(x)\|})$, normalizing over all the classes. Finally, we can compute the prototype-based classification loss for $\mathcal{D}_{\mathcal{S}}$ with standard cross-entropy loss:

$$\mathcal{L}_{SC} = \mathbb{E}_{(x,y) \in \mathcal{D}_S} \mathcal{L}_{ce}(h(x), y) \tag{1}$$

The intuition behind this loss is that the higher the confidence of sample $x$ being classified as class $i$, the closer the embedding of $x$ is to $\mathbf{w}_i$. Hence, when optimizing Equation 1, we are reducing the intra-class variation by pushing the embedding of each sample $x$ closer to its corresponding weight vector in $\mathbf{W}$. In this way, $\mathbf{w}_i$ can be seen as a representative data point (prototype) for $p(x|y = i)$.

**Prototype-based Conditional Alignment by Minimax Entropy** Due to the lack of categorical information in the target domain, it is infeasible to utilize Equation 1 to obtain target prototypes. Instead, we propose to tackle this problem by 1) moving each source prototype to be closer to its nearby target samples, and 2) clustering target samples around this moved prototype. Inspired by Saito et al. (2019a), we propose to achieve these two objectives jointly by entropy minimax learning. Specifically, for each sample $x^t \in \mathcal{D}_{\mathcal{T}}$ fed into the network, we can compute the entropy of the classifier's output by

$$H = -\mathbb{E}_{x \in \mathcal{D}_{\mathcal{T}}} \sum_{i=1}^{c} h_i(x) \log h_i(x). \tag{2}$$

Larger $H$ indicates the target samples are similar to all the weight vectors (prototypes) of $C$. We achieve conditional feature distributions alignment by aligning the source and target prototypes in an adversarial process: (1) we train $C$ to *maximize $H$*, aiming to move the prototypes from the source samples towards the neighboring target samples; (2) we train $F$ to *minimize $H$*, aiming to make the embedding of target samples closer to their nearby prototypes. By training with these two objectives as a min-max game between $C$ and $F$, we can align source and target prototypes. Practically, we add a gradient-reverse layer (Ganin & Lempitsky, 2015a) between $C$ and $F$ to flip the sign of gradient.

---

**Algorithm 1:** Conditional Feature Alignment with Decision Boundary Refinement

---

**Input:** $F$, $C$, $\mathcal{D}_S$, $\mathcal{D}_T$, $N_{iter}$, $N_{epoch}$, $N_b$, $k = k_0$
**Output:** $\hat{F}$, $\hat{C}$

1   train $F$ and $C$ with $\mathcal{D}_S$ using Equation 1
2   **for** $i = 0, ..., N_{epoch}$ **do**
3      **Step A**:
4        compute $\hat{\mathcal{D}}_T = \{(x_i^t, \hat{y}_i^t, m_i)_{i=1}^{N_t}\}$ with $k$ as in Section 3.2
5      **Step B**:
6      **for** $j = 0, ..., N_{iter}$ **do**
7        randomly sample mini-batch $\{(x_i^s, y_i^s)_{i=1}^{N_b}\} \in \mathcal{D}_S$, $\{(x_i^t, \hat{y^t}_i, m_i)_{i=1}^{N_b}\} \in \hat{\mathcal{D}}_T$
8        train $F$ and $C$ with Equation 4
9      $k = \min(k + k_{step}, k_{max})$

---

## 3.2   Label Distribution Estimation with Self-training

As the source label distribution $p(y)$ is different from that of the target $q(y)$, it is not guaranteed that the classifier $C$ which has low risk on $\mathcal{D}_S$ will have low error on the target domain. Intuitively, if the classifier is trained with imbalanced source data, the decision boundary will be dominated by the most frequent categories in the training data, leading to a "biased" classifier. When the classifier is applied to target domain with a different label distribution, its accuracy will degrade as it is highly biased. To tackle this problem, we employ the *self-training* method to estimate the target label distribution and refine the decision boundary. In addition, we leverage *balanced sampling* of the source data to further diminish the effect of label shift.

**Self-training** In order to refine the decision boundary, we propose to estimate the target label distribution with self-training (Zou et al., 2018). We will assign pseudo labels $\hat{y}$ to all the target samples according to the output the classifier $C$. As we have aligned the conditional feature distributions ($p(x|y)$ and $q(x|y)$), we assume that the pseudo label distribution $q(\hat{y})$ is the approximation of the real label distribution $q(y)$ for the target domain.

For each category, we select top-$k$ percent of the target samples with the highest confidence scores belonging to that category. We utilize the highest probability in $h(x)$ as the classifier's confidence on sample $x$. Specifically, for each pseudo-labeled sample $(x, \hat{y})$, we set its selection mask $m = 1$ if $h(x)$ is among the top-$k$ percent of all the target samples with the same pseduo-label, otherwise $m = 0$. Denote the pseudo-labeled target set as $\hat{\mathcal{D}}_T = \{(x_i^t, \hat{y}_i^t, m_i)_{i=1}^{N_t}\}$, we leverage the data and pseudo labels from $\hat{\mathcal{D}}_T$ to train the classifier $C$, aiming to refine the decision boundary with target label distribution. The total loss function for classification is:

$$\mathcal{L}_{ST} = \mathcal{L}_{SC} + \mathbb{E}_{(x,\hat{y},m) \in \hat{\mathcal{D}}_T} \mathcal{L}_{ce}(h(x), \hat{y}) \cdot m \tag{3}$$

where $\hat{y}$ indicates the pseudo labels and $m$ indicates selection masks. In our approach, we choose the top-k percent of the highest confidence target samples within each category, instead of universally. This is crucial to estimate the real target label distribution, otherwise, the easy-to-transfer category(ies) will dominate $\hat{\mathcal{D}}_T$, leading to inaccurate estimation of the target label distribution.

**Balanced Sampling of Source Data** In addition to estimating the target label distribution, we propose a balanced sampling process to enhance label shift alignment. When coping with label shift, the label distribution of the source domain could be highly unbalanced. A classifier trained on unbalanced categories will make highly-biased predictions for the samples from the target domain (He & Garcia, 2009). This effect also hinders the self-training process discussed above, as the label distribution estimation will also be biased. To tackle these problems, we apply a balanced mini-batch sampler to generate training data from the source domain and ensure that each source mini-batch contains roughly the same number of samples for each category.

## 3.3   Training Process

In this section, we combine the above ideas into an end-to-end training process. Given input samples from source domain $\mathcal{D}_S$ and target domain $\mathcal{D}_T$, we first initialize our network $F$ and $C$ with only

labeled data $\mathcal{D}_S$. Then, we iterate between the assignment of pseudo labels and adaptive learning until convergence or reaching the maximum number of iterations. An overview of this process can be seen in Algorithm 1. The overall training objective is as follows:

$$
\begin{aligned}
\hat{C} &= \arg\min_C \mathcal{L}_{ST} - \alpha H \\
\hat{F} &= \arg\min_F \mathcal{L}_{ST} + \alpha H
\end{aligned}
\tag{4}
$$

### 3.4 THEORETICAL INSIGHTS

**Conditional Feature Alignment** According to Zhao et al. (2019b), the target error in domain adaptation is bounded by three terms: 1) source error, 2) the discrepancy between the marginal distributions and 3) the distance between the source and target optimal labeling functions. Denote $h \in \mathcal{H}$ as the hypothesis, $\epsilon_S(\cdot)$ and $\epsilon_T(\cdot)$ as the expected error of a labeling function on source and target domain, and $f_S$ and $f_T$ as the optimal labeling functions in the source and target domain. Then, we have:

$$
\epsilon_T(h) \leq \epsilon_S(h) + d_{\hat{\mathcal{H}}}(\mathcal{D}_S, \mathcal{D}_T) + \min\{\epsilon_S(f_T), \epsilon_T(f_S)\},
\tag{5}
$$

where $d_{\hat{\mathcal{H}}}$ denote the discrepancy of the marginal distributions (Zhao et al., 2019b). The bound demonstrates that the optimal labeling functions $f_S$ and $f_T$ need to generalize well in both domains, such that the term $\min\{\epsilon_S(f_T), \epsilon_T(f_S)\}$ can be bounded. Conventional domain adaptation approaches which only align marginal feature distribution cannot guarantee that $\min\{\epsilon_S(f_T), \epsilon_T(f_S)\}$ is minimized. This motivates us to align the conditional feature distribution, *i.e.* $p(x|y)$ and $q(x|y)$. Our model COALestimates and aligns the source and target prototypes to align $p(x|y)$ and $q(x|y)$.

**Decision Boundary Refinement** Theorem 4.3 in Zhao et al. (2019b) indicates that the target error $\epsilon_T(h)$ can not be minimized if we only align the feature distributions and neglect the shift in label distribution. Denote $d_{JS}$ as the Jensen-Shannon(JS) distance between two distributions, Zhao et al. (2019b) propose:

$$
\epsilon_S(h) + \epsilon_T(h) \geq \frac{1}{2}(d_{JS}(p(y), q(y)) - d_{JS}(p(x), q(x)))^2
\tag{6}
$$

This theorem demonstrates that when the divergence between label distributions $d_{JS}(p(y), q(y))$ is significant, minimizing the divergence between marginal distributions $d_{JS}(p(x), q(x))$ and the source task error $\epsilon_S(h)$ will enlarge the target task error $\epsilon_T(h)$. Motivated by this, we propose to align the empirical label distributions with self-training algorithm.

## 4 EXPERIMENTS

We evaluate our CO-ALignment (**COAL**) model with three popular benchmarks, *i.e.*, **Digits**, **Office-Home** (Venkateswara et al., 2017) and **DomainNet** (Peng et al., 2019a). Sample images of these datasets can be found in Figure 3a. As Digits and Office-Home dataset contains limited SLD, we artificially create considerable SLD by random sub-sampling the data for each category.

### 4.1 SETUP

**Sub-sampling Protocol.** To create significant SLD between source and target domains, we sub-sample the current datasets with **R**eversely-unbalanced **S**ource and **U**nbalanced **T**arget (**RS-UT**) protocol. In this setting, both the source and target domains have unbalanced label distribution, while the label distribution of the source domain is a reversed version of that of the target domain. An illustration of this setting can be found in Figure 3(a). We refer our reader to supplementary material for more details!

**Digits** We select four digits datasets: MNIST (LeCun et al., 1998), USPS (Hull, 1994), SVHN (Netzer et al., 2011) and Synthetic Digits (SYN) (Ganin & Lempitsky, 2015b). In this work, we investigate four domain adaptation tasks: **MNIST → USPS**, **USPS → MNIST**, **SVHN → MNIST**, and **SYN → MNIST**.

**Office-Home** (Venkateswara et al., 2017) is a dataset collected in office and home environment with 65 object classes and four domains: Real World (**Rw**), Clipart (**Cl**), Product (**Pr**), Art (**Ar**). Since

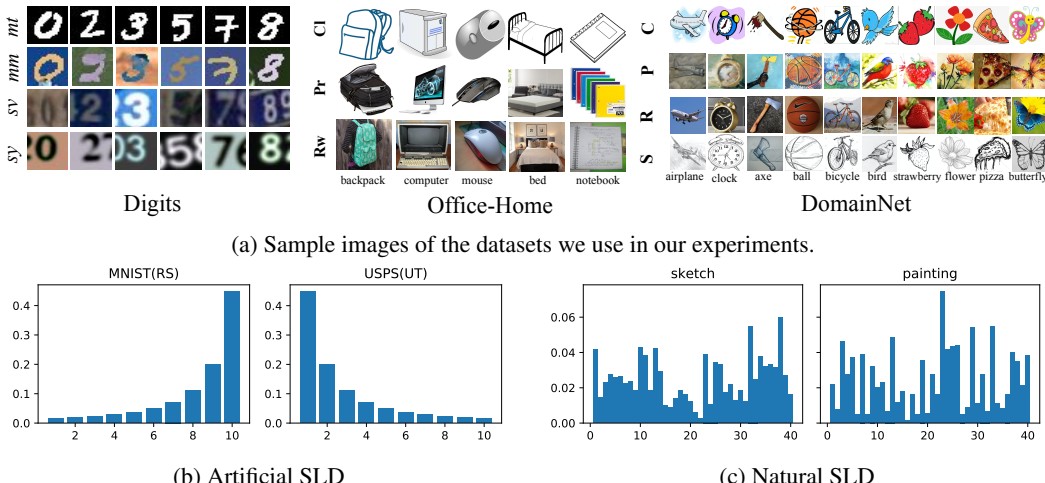

(a) Sample images of the datasets we use in our experiments.

(b) Artificial SLD                    (c) Natural SLD

Figure 3: (**a**): Image examples from Digits, Office-Home (Venkateswara et al., 2017), and Domain-Net Peng et al. (2019a). (**b**): illustrations of **R**eversely-unbalanced **S**ource (**RS**) and Unbalanced Target (**UT**) distribution in MNIST→USPS task. (**c**): Natural SLD of DomainNet.

| Methods | USPS→MNIST | MNIST→USPS | SVHN→MNIST | SYN →MNIST | AVG |
|---|---|---|---|---|---|
| Source Only | 75.31±0.09 | 87.92±0.74 | 50.25±0.81 | 85.74±0.49 | 74.81 |
| F-DANN | 72.59±1.61 | 81.62±2.38 | 45.65±2.93 | 82.07±1.65 | 70.48 |
| JAN | 75.75±0.75 | 78.82±0.93 | 53.21±3.94 | 75.64±1.42 | 70.86 |
| BBSE | 75.01±3.68 | 78.84±10.73 | 49.01±2.02 | 85.69±0.71 | 72.14 |
| BSP | 71.99±1.52 | 89.74±0.77 | 50.61±1.67 | 77.30±1.20 | 72.41 |
| PADA | 73.66±0.15 | 78.59±0.23 | 54.13±1.61 | 85.06±0.60 | 72.86 |
| MCD | 77.18±5.65 | 85.34±4.07 | 53.52±4.23 | 76.37±3.48 | 73.10 |
| DAN | 79.12±1.34 | 87.15±1.71 | 53.63±1.80 | 80.89±2.00 | 75.20 |
| DANN | 77.28±2.13 | 91.88±0.74 | 57.16±1.83 | 77.60±1.29 | 75.98 |
| **COAL** (Ours) | **88.12±0.37** | **93.04±1.67** | **65.67±1.29** | **90.60±0.44** | **84.33** |

Table 1: Per-class mean accuracy on Digits with RS-UT label shifts. Our model achieves **84.33**% average accuracy across four experimental settings, outperforming other baselines.

the "Art" domain is too small to sample an imbalanced subset, we focus on the remaining domains and explore all the six adaptation scenarios.

**DomainNet** (Peng et al., 2019a) is a large-scale testbed for domain adaptation, which contains six domains with about 0.6 million images distributed among 345 classes. Since some domains and classes contains many outliers, we select 40 commonly-seen classes from four domains: Real (**R**), Clipart (**C**), Painting (**P**), Sketch (**S**). Different from two datasets above, the SLD in DomainNet is significant enough, as illustrated in Figure 3. Moreover, the images are directly collected from image search engine and they are in consistence with the natural label distribution in the real world.

**Baselines** We compare covarite and SLD **CO-AL**ignment (**COAL**) with state-of-the-art domain adaptation methods that mainly aligns feature distribution: Deep Adaptation Network (**DAN**) (Long et al., 2015), Joint Adaptation Network (**JAN**) (Long et al., 2017), Domain Adversarial Neural Network (**DANN**) (Ganin & Lempitsky, 2015a), Batch Spectral Penalization (**BSP**) (Chen et al., 2019c), f-divergence Domain Adversarial Neural Network (**F-DANN**) (Wu et al., 2019), Maximum Classifier Discrepancy (**MCD**) (Saito et al., 2018). We also compare our model with methods that aligns the marginal label distribution: Partial Adversarial Domain Adaptation (**PADA**) (Cao et al., 2018a), Black Box Shift Estimation (**BBSE**) (Lipton et al., 2018b).

**Implementation Details.** We implement all our experiments in Pytorch[1] platform. For the Digits dataset, we adopt the network architecture proposed by Saito et al. (2018). We adopt SGD with the momentum of 0.9 and learning rate of 0.01 for the linear classifier and 0.001 for all other layers. The batch size is set as 32 for samples from each domain. For the other two datasets, we utilize ResNet-

---

[1]https://pytorch.org/

| Methods | Rw→Pr | Rw→Cl | Pr→Rw | Pr→Cl | Cl→Rw | Cl→Pr | AVG |
|---|---|---|---|---|---|---|---|
| Source Only | 69.77 | 38.35 | 67.31 | 35.84 | 53.31 | 52.27 | 52.81 |
| BSP | 72.80 | 23.82 | 66.19 | 20.05 | 32.59 | 30.36 | 40.97 |
| PADA | 60.77 | 32.28 | 57.09 | 26.76 | 40.71 | 38.34 | 42.66 |
| BBSE | 61.10 | 33.27 | 62.66 | 31.15 | 39.70 | 38.08 | 44.33 |
| MCD | 66.03 | 33.17 | 62.95 | 29.99 | 44.47 | 39.01 | 45.94 |
| DAN | 69.35 | 40.84 | 66.93 | 34.66 | 53.55 | 52.09 | 52.90 |
| F-DANN | 68.56 | 40.57 | 67.32 | 37.33 | 55.84 | 53.67 | 53.88 |
| JAN | 67.20 | 43.60 | 68.87 | 39.21 | 57.98 | 48.57 | 54.24 |
| DANN | 71.62 | **46.51** | 68.40 | 38.07 | 58.83 | **58.05** | 56.91 |
| **COAL** (Ours) | **73.65** | 42.58 | **73.26** | **40.61** | 59.22 | 57.33 | **58.40** |

Table 2: Per-class mean accuracy on Office-Home dataset with RS-UT label shifts. Our model achieve **58.40**% average accuracy across six GDA tasks, outperforming other baselines.

| Methods | R→C | R→P | R→S | C→R | C→P | C→S | P→R | P→C | P→S | S→R | S→C | S→P | AVG |
|---|---|---|---|---|---|---|---|---|---|---|---|---|---|
| Source Only | 58.84 | 67.89 | 53.08 | 76.70 | 53.55 | 53.06 | 84.39 | 55.55 | 60.19 | 74.62 | 54.60 | 57.78 | 62.52 |
| BBSE | 55.38 | 63.62 | 47.44 | 64.58 | 42.18 | 42.36 | 81.55 | 49.04 | 54.10 | 68.54 | 48.19 | 46.07 | 55.25 |
| PADA | 65.91 | 67.13 | 58.43 | 74.69 | 53.09 | 52.86 | 79.84 | 59.33 | 57.87 | 76.52 | 66.97 | 61.08 | 64.48 |
| DAN | 64.36 | 70.65 | 58.44 | 79.44 | 56.78 | 60.05 | 84.56 | 61.62 | 62.21 | 79.69 | 65.01 | 62.04 | 67.07 |
| MCD | 61.97 | 69.33 | **79.78** | 79.78 | 56.61 | 53.66 | 83.38 | 58.31 | 60.98 | 81.74 | 56.27 | 66.78 | 67.38 |
| FDANN | 66.15 | 71.80 | 61.53 | 81.85 | 60.06 | 61.22 | 84.46 | 66.81 | 62.84 | 81.38 | 69.62 | 66.50 | 69.52 |
| JAN | 65.57 | 73.58 | 67.61 | 85.02 | 64.96 | 67.17 | 87.06 | 67.92 | 66.10 | 84.54 | 72.77 | 67.51 | 72.48 |
| BSP | 67.29 | 73.47 | 69.31 | **86.50** | 67.52 | 70.90 | 86.83 | 70.33 | 68.75 | 84.34 | 72.40 | 71.47 | 74.09 |
| DANN | 63.37 | 73.56 | 72.63 | 86.47 | 65.73 | 70.58 | 86.94 | **73.19** | **70.15** | **85.73** | **75.16** | 70.04 | 74.46 |
| **COAL** (Ours) | **73.85** | **74.82** | 70.50 | 85.68 | **70.86** | **71.29** | **87.71** | 69.90 | 67.28 | 83.82 | 73.21 | **70.53** | **74.96** |

Table 3: Per-class mean accuracy on DomainNet dataset with natural label shifts. Our method achieve **74.96**% average accuracy across the 12 experiments. We observe that our model has significant performance boost on several tasks, such as **R→C**, and **C→P**. Note that DomainNet contains about 0.6 million images, it is non-trivial to have even one percent performance boost.

50 (He et al., 2015) as our backbone network, and replace the last fully-connected layer with a randomly initialized N-way classifier layer (for N categories). We also use SGD with momentum of 0.9 while setting the learning rate to be 0.001 for linear layers and 0.0001 for all the other layers. The batch size is set as 16 for each domain. We set the $\alpha = 1$ in Equation 4 and set $k_0 = 5, k_{step} = 5, k_{max} = 50$ as the parameters for self-training selection policy. For all the compared methods, we decide their hyper-parameters on the validation set of Painting → Clipart task in DomainNet.

**Evaluation metric.** When the target domain is highly unbalanced, conventional overall average accuracy that treats every class uniformly is not an appropriate performance metric (He & Garcia, 2008). Therefore, we follow Dong et al. (2019) to use the *per-class* mean accuracy in our main results. Formally, we denote $S_i = \frac{n_{(i,i)}}{n_i}$ as the accuracy for class $i$, where $n_{(i,j)}$ represents the number of class $i$ samples labeled as class $j$, and $n_i = \sum_{j=1}^{c} n_{(i,j)}$ represents the number of samples in class $i$. Then, the per-class mean accuracy is computed as $S = \frac{1}{c} \sum_{i=1}^{c} S_i$.

## 4.2 RESULTS

We first show the experimental results of our model and the compared baselines on Digits dataset in Table 1. The covariate shift between the source and target domains is related small on Digits dataset. For fair comparison, we leverage the same backbones for all the models. From the results, we can make the following observations. (1) Our model achieves **84.33**% average accuracy across four experimental setting, outperforming the best-performing baselines by **8.4**%! (2) Within the eight selected baselines, only **DAN** and **DANN** have marginal improvement (less than 2%) over the source-only baseline. The other six compared methods encounter the *negative transfer* (Pan & Yang, 2010) issue and perform worse than the source-only baselines. These results demonstrate that aligning only the marginal feature distributions or only the label distributions can not fully tackle GDA task. In contrast, our model co-aligns the conditional feature distributions and label distributions and outperforms the baselines by a large margin.

Next, we show the experimental results on more challenging real-image datasets, *i.e.* Office-Home and DomainNet, in Table 2 and Table 3, respectively. In Office-Home experiments, our model

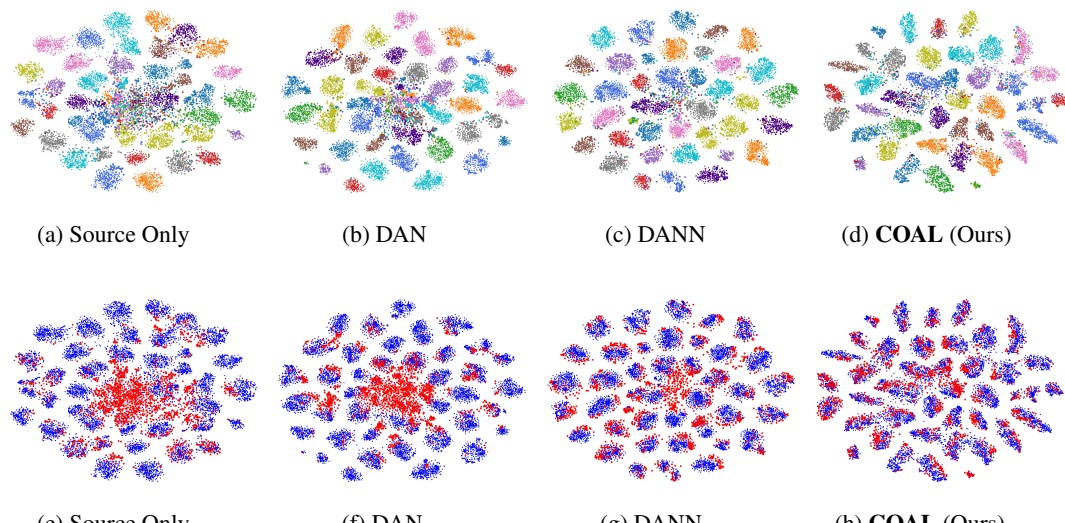

(a) Source Only      (b) DAN      (c) DANN      (d) **COAL** (Ours)

(e) Source Only      (f) DAN      (g) DANN      (h) **COAL** (Ours)

Figure 4: Feature visualization: t-SNE plot for source features, DAN features, DANN features, and COAL features on GDA task Real → Clipart. Figure (a)-(d): features from each class. Different colors denote different categories. Figure (e)-(h): features from each domain. Blue and red points represents features from the source domain and target domain, respectively.

achieves **58.40**% average accuracy across the six GDA tasks, outperforming other compared baselines. Our model has **5.59**% improvement from the source-only result. We also notice that the **BSP**, **PADA**, **BBSE** and **MCD** models perform worse than the source-only baselines, which is in consistent with the results on Digits dataset. The empirical results on Office-Home dataset demonstrate the effectiveness of co-alignment of conditional covariate shift and SLD.

In DomainNet experiments, our model get **74.96**% average accuracy across the 12 experiments, outperforming the compared baselines. Note we have carefully tuned the hyper-parameters for the compared domain adaptation methods. Without the hyper-parameters tuning, *i.e.* directly applying the model released by the authors, these models perform worse than the source-only baselines. In addition, our model improves the performance of source-only models by **12.4**%.

## 4.3 ANALYSIS

**Ablation Study.** Our COAL method has mainly two objectives: 1) alignment of conditional feature distribution and 2) alignment of label distribution. To show the importance of these two objectives in GDA, we show the performance of our method without each of these objectives respectively on multiple tasks. The results in Table 4 showed the importance of both objectives. For example, for task USPS → MNIST, if we remove the conditional feature distribution alignment objective, the accuracy of our model will drop by 2.55%. Similarly, if we remove the label distribution alignment objective, the accuracy will drop by 2.9%. These results demonstrate that both the alignment of conditional feature distribution and label distribution are important to GDA task.

**Feature Visualization.** In this section, we plot the learned features with t-SNE (van der Maaten & Hinton, 2008) in Figure 4. We investigate the Real to Clipart task in DomainNet experiment with ResNet-50 backbones. From (a)-(d), we can observe that our method has better ability to cluster each sample towards its class centroid. This confirms the effectiveness of the entropy minimization in Equation 4. From (e)-(h), we observe that our method can better align source and target features in each category, while other methods either leave the feature distributions unaligned, or incorrectly aligned samples in different categories. These results further show the importance of *prototype-based conditional feature alignment* for generalized domain adaptation task.

**Effect of Source Balanced Sampler.** Source balanced samplers described in Section 3.2 can help us tackle the bias-classifier problem caused by the imbalanced data distribution of source domain. A significant performance boost can be observed after applying the balanced sampler for our COAL model, as well as the compared baselines. In this section, we specifically show the effect of using source balanced samplers. We show in Table 5 the performance of several methods with and without source balanced samplers on 5 adaptation tasks from multiple datasets. We observe that for 22 of the total 25 tasks (5 models on 5 adaptation tasks), using source balanced samplers

| Methods | U→M | M→U | Syn→M | Cl→Rw | Pr→Rw | R→C | P→C | P→R | AVG |
|---------|-----|-----|-------|-------|-------|-----|-----|-----|-----|
| w/o self-training | 85.22 | 85.94 | 55.17 | 58.38 | 69.39 | 71.92 | 62.86 | 77.45 | 70.79 |
| w/o conditional alignment | 85.57 | 92.28 | 63.34 | 59.41 | 72.11 | 71.34 | 68.50 | 87.14 | 74.96 |
| **COAL** (full model) | 88.12 | 93.04 | 65.67 | 59.81 | 73.46 | 73.65 | 69.90 | 87.71 | 76.42 |

Table 4: Ablation study of different objectives in our method. We randomly select 8 sets of experiments to perform the ablation study. The results demonstrate that the *prototype-based conditional alignment* and *self-training* are both critical for GDA.

| Methods | USPS→MNIST | | SVHN→MNIST | | Pr→Cl | | Cl→Rw | | R→S | |
|---------|------|------|------|------|------|------|------|------|------|------|
|         | w/o  | with | w/o  | with | w/o  | with | w/o  | with | w/o  | with |
| SourceOnly | 71.35 | **74.98** | 50.35 | 50.25 | 34.99 | **35.84** | 50.64 | **53.31** | 50.16 | **53.08** |
| DAN | 64.81 | **79.37** | 22.05 | **48.19** | 32.93 | **34.66** | 45.18 | **53.55** | 64.78 | 58.44 |
| DANN | 42.77 | **77.48** | 27.60 | **56.35** | 35.17 | **38.07** | 47.19 | **58.05** | 68.92 | **72.63** |
| MCD | 20.15 | **79.59** | 44.83 | **51.91** | 33.06 | 29.99 | 49.57 | 44.47 | 58.50 | **79.78** |
| COAL | 87.50 | **88.12** | 60.12 | **65.67** | 34.03 | **40.61** | 57.67 | **59.22** | 59.23 | **70.50** |

Table 5: The performance of five models *w.* or *w/o.* source balanced sampler. We observe a significant performance boost when the source balanced sampler is applied, both for our model and the compared baselines, demonstrating the effectiveness of source balanced sampler to GDA task.

will improve the domain adaptation performance. Furthermore, in USPS → MNIST and SVHN → MNIST, source balance samplers can prevent the baseline methods from *negative transfer*. These results show the effectiveness of having a source balanced sampler when tackling GDA task.

**Different Degrees of SLD.** In Section 4, we only investigated the performance of each method under a certain kind of SLD in each dataset, *i.e.*, RS-UT. In this section, we investigate the effect of different degrees of SLD on the performance of domain adaptation methods. Specifically, we create 4 interval degrees of SLD between the **BS-BT** (**B**lanced **S**ource and **B**lanced **T**arget) and RS-UT setting. To this end, we compute the proportions of each category by linear interpolation between its proportions in BS-BT and RS-UT. We denote BS-BT and RS-UT as $0\%$ and $100\%$ degree of SLD respectively, and create datasets with SLDs of $20\%, 40\%, 60\%$ and $80\%$ on the linear interpolation. For fair comparison, all the datasets have the same total amount of samples. With these datasets, we evaluated the performances of different methods on the USPS → MNIST task of Digits. The results in Figure 5 show that the performances of previous domain adaptation methods will be significantly affected by SLD. For

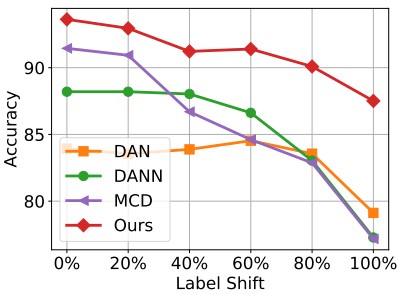

Figure 5: Performance on USPS→MNIST task with different degrees of SLD. $0\%$ and $100\%$ denote the BS-BT and RS-UT settings respectively. Others are the linear interpolations of BS-BT and RS-UT. The results demonstrate that our model are more robust to different degrees of SLD.

example, the accuracy of MCD drastically drops from **91.45**% to **77.18**%. In contrast, the performance our method is much more stable, which ranges between **93.42**% and **88.12**%. It shows that our method are robust to different degrees of SLD.

## 5 CONCLUSION

In this paper, we first propose a generalized domain adaptation (**GDA**) learning schema and demonstrate the importance of GDA in practical scenarios. Then we deliver theoretically analysis to demonstrate that aligning conditional feature distributions ($p(x|y)$, $q(x|y)$) and aligning the label distributions ($p(y)$,$q(y)$) are essential for GDA. Towards tackling GDA task, we have proposed conditional feature distribution and label distribution CO-ALignment (**COAL**) approach, which leverages *prototype-based conditional alignment* and *self-training* method. Empirically, we demonstrate that previous UDA models which learn domain-invariant features by aligning the marginal distributions fail to tackle GDA task, due to the presence of shift in the label distribution. An extensive empirical evaluation on GDA benchmarks demonstrate the efficacy of the proposed model against several state-of-the-art domain adaptation algorithms.

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

## A  ADDITIONAL RESULTS

We have shown in the paper the performance of different domain adaptation methods when conditional covariant shift and label shift exist simultaneously. To analysis the influence of label shift in unsupervised domain adaptation, in this appendix we also include the results on datasets without label shift. To this end, we also created datasets with the Balanced Source Balanced Target (BS-BT) setting. Specifically, under BS-BT every class has the same number of samples. We use this setting to sub-sample the BS-BT datasets from Digits and Office-Home. For fair comparison with the results in Section 4, we constrain that the total amount of samples of each domain is the same in BS-BT and RS-UT. We show the results in Table 1, and 2.

| Methods | USPS→MNIST | MNIST→USPS | SVHN→MNIST | SynD→SVHN | AVG |
|---------|-----------|-----------|-----------|-----------|-----|
| Source Only | 71.49±0.08 | 84.73±1.34 | 56.95±1.52 | 86.33±0.56 | 76.00 |
| FDANN | 72.13±1.88 | 87.30±0.99 | 34.16±1.64 | 80.77±1.45 | 67.41 |
| BBSE | 79.42±3.04 | 74.07±3.56 | 53.73±3.19 | 87.14±0.88 | 71.65 |
| PADA | 73.07±0.13 | 81.39±0.11 | 56.82±0.97 | 85.88±0.35 | 74.70 |
| JAN | 87.69±0.51 | 93.93±0.34 | 56.00±4.30 | 82.65±3.02 | 77.53 |
| DAN | 83.09±2.10 | 86.16±0.49 | 63.42±2.08 | 86.48±0.97 | 78.69 |
| BSP | 83.33±1.77 | 89.21±1.44 | 63.31±2.43 | 93.24±0.53 | 81.92 |
| DANN | 83.92±1.19 | 90.29±0.94 | 68.96±3.73 | 93.43±0.36 | 84.23 |
| MCD | **97.53±0.04** | **96.86±0.60** | **88.47±1.96** | **96.82±0.32** | **94.92** |
| COAL (Ours) | 92.87±0.60 | 95.40±0.59 | 82.01±2.36 | 96.39±0.26 | 91.27 |

Table 1: Class-balanced accuracy on Digit dataset with BS-BT label shifts

| Methods | Rw→Pr | Rw→Cl | Pr→Rw | Pr→Cl | Cl→Rw | Cl→Pr | AVG |
|---------|-------|-------|-------|-------|-------|-------|-----|
| Source Only | 72.91 | 39.21 | 69.84 | 38.16 | 56.06 | 54.08 | 55.04 |
| PADA | 64.44 | 33.89 | 59.21 | 30.35 | 41.92 | 41.19 | 45.17 |
| BBSE | 64.25 | 34.85 | 63.56 | 34.19 | 40.57 | 38.57 | 46.00 |
| BSP | 72.85 | **50.29** | 69.35 | 25.20 | 34.31 | 31.92 | 47.32 |
| MCD | 68.76 | 37.14 | 66.78 | 36.31 | 51.77 | 48.50 | 51.54 |
| DAN | 71.67 | 38.35 | 67.42 | 36.07 | 55.48 | 53.25 | 53.71 |
| FDANN | 71.20 | 39.11 | 67.30 | 33.86 | 60.85 | 53.64 | 54.33 |
| JAN | 66.69 | 42.28 | 69.61 | 38.65 | 59.42 | 57.19 | 55.64 |
| DANN | 74.70 | 47.10 | 70.07 | 39.35 | 60.56 | 54.88 | 57.78 |
| COAL | **77.44** | 47.42 | **71.23** | **42.87** | **63.46** | **61.24** | **60.61** |

Table 2: Class-balanced accuracy on Office-Home dataset with BS-BT label shifts

We can observe that conventional methods that focus on covariate shift can achieve decent improvement over the Source Only baseline in BS-BT setting. For example, MCD is largely improving the performance when there is no label shift between the source and target domain. However, as we observed in Table 1 of our paper, MCD is actually having negative transfer effect when applied to the more general GDA setting. This comparison demonstrates the importance of considering both covariate and label shift in our generalized setting. Please note that the results in this section are produced by models with hyper-parameters tuned on unbalanced DomainNet.

## B  CREATION DETAILS FOR LABEL SHIFT

In order to create unbalanced label distribution in each dataset, inspired by Liu et al. (2019), we follow the Paredo distribution set different proportions for each category. By using this distribution, we can create *long-tailed* label distribution, which is frequently seen in real applications and benchmarks (Van Horn et al., 2018; Peng et al., 2019a).

The shape of Paredo distribution (Reed, 2001) is controlled by parameter $\alpha$. Because different datasets have different amount of samples, to avoid making some classes in the unbalanced dataset to have too few samples, we use a different parameter for each dataset. Specifically, we set $\alpha = 1$ for Digits dataset, and $\alpha = 100$ for Office-Home dataset.

We further assign each computed proportion to each category by following the descending order of the original class index. Specifically, in the target domain of RS-UT, we assign the $k_{th}$ largest propotion to class $k - 1$, with class index starting from 0. For Digits, we set the index of each class directly as the digit it represents. For Office-Home, we set the class index in alphabetical ascending order.

## C    IMPLEMENTATION DETAILS FOR COMPARED METHODS

In our experiments, we use the authors' official implementations for DAN (Long et al., 2015), and JAN (Long et al., 2017), PADA (Cao et al., 2018b), MCD (Saito et al., 2018), and BSP (Chen et al., 2019c). For DANN (Ganin & Lempitsky, 2015a), BBSE (Lipton et al., 2018b) and FDANN (Wu et al., 2019), we implement their method by ourselves.

We tune the hyper-parameters of each method on Painting $\rightarrow$ Clipart task in DomainNet. Specifically, for PADA, BSP, DAN, JAN and DANN we tune the weight $\alpha$ of the marginal feature alignment loss. We empirically find that these method achieve better performance when we set $\alpha$ to be 5 10 times *lower* than default. Intuitively, it means that we can achieve better performance under generalized domain shift setting if we relax the strength of marginal feature alignment. For MCD we tune the number of feature generator updating times $n$.

## D    DETAILED INFORMATION FOR DATASETS

We provide detailed information for datasets in Table 3.

| | Digits | | | | |
|---|---|---|---|---|---|
| Splits | USPS | MNIST | SVHN | SYN | Total |
| Train | 12,144 | 1,550 | 10,395 | 107,005 | 118,950 |
| Test | 2,181 | 459 | 3,554 | 2,114 | 8,308 |
| | Office-Home | | | | |
| Splits | Real World | Product | Clipart | | Total |
| Total | 1,253 | 2,045 | 1,017 | | 4,315 |
| | DomainNet | | | | |
| Splits | Real | Painting | Clipart | Sketch | Total |
| Train | 16,141 | 6,727 | 3,707 | 5,537 | 32,112 |
| Test | 6,943 | 2,909 | 1,616 | 2,399 | 13,867 |

Table 3: Detailed information for datasets

