# OpenReview forum: "Generalized Domain Adaptation with Covariate and Label Shift CO-ALignment"
_ICLR.cc/2020/Conference — Reject_

### Official Review · AnonReviewer2 · 2019-10-23
**Official Blind Review #2**

**Rating:** 6

**Review:**

The paper deals with covariate and label shift in common
evaluated on some standard benchmark data.
The paper is scientific sound and appears to be in good shape,
just a few comments:
- how is your label shift different to concept drift in supervised learning?
  --> if this is basically the same I would expect that you can use,mention methods from there
- the plots with t-SNE are obviously colorful but do not provide a lot of information - they should
  be removed - I do not see a benefit
- it is very common that all these methods (like yours) are provided on some kind of image
  data ... is there a particular reason / limitation?
- I would like to see additional experiments on similar text data -> reuters
- self-training is a concept from semi-supervised learning and not particular well supported
  in the community - how do you make sure that the result remain valid
- how do you make sure that the adaptation of the labels, the covariate shift and the classifier training
  are not in facting cheating the result to an optimum within the optimized cost function?


**Experience Assessment:**

I have published one or two papers in this area.

**Review Assessment: Checking Correctness Of Derivations And Theory:**

I assessed the sensibility of the derivations and theory.

**Review Assessment: Checking Correctness Of Experiments:**

I assessed the sensibility of the experiments.

**Review Assessment: Thoroughness In Paper Reading:**

I read the paper at least twice and used my best judgement in assessing the paper.

---

> ### Author Response · Authors · 2019-11-15
> **Response to Reviewer #2**
>
> Reviewer #2
>
> Thank you for your time and effort! We truly appreciate your recognition of the strength of our paper.
>
> Here, we address your comments as below:
>
> Q1. How is your label shift different to concept drift in supervised learning?
> The label shift mentioned in our paper refers to the fact that the label distributions of the source and target domains are different ($p(y) \not= q(y)$). Concept drift on the other hand focuses on the problem that $p(y|x) \not= q(y|x)$.
>
> Q2. Why the experiments are conducted only on image data?
> Thank you for your suggestions. Currently, our problem and approach are targeted at visual domain adaptation, therefore, we did not provide experiments on non-image datasets. However, we think it's worth trying other non-image datasets with a shift in feature distribution and label distribution simultaneously. We leave the exploration of non-image dataset using our model as future work.
>
> Q3. How do you make sure the result of self-training remains valid?
>  To make this process more robust, we use the confidence score to select the most reliable pseudo labels. We empirically found this process is effective in multiple domain pairs and across multiple domain adaptation datasets, which encourages us to incorporate it into our method. Improving the robustness of self-training in our context could be a very interesting and important path to enhance our method.
>
> Q4. How to make sure the performance is not the fact of cheating the result to a local optimum with the optimized cost function?
> Our loss function is composed of the self-training classification loss for aligning label distributions and the minimax entropy loss for aligning conditional feature distributions. How to deal with the competition of these two losses during optimization is a long-lasting problem in deep learning research, which is beyond the scope of this paper. As we empirically demonstrated, most of the previous unsupervised deep domain adaptation methods fail to work on the proposed Generalized Domain Adaptation problem. Therefore, the goal of our method is to effectively tackle the domain adaptation with the shift in feature distribution and label distribution simultaneously, not to study the strategy to optimize the proposed loss functions to a global optimum.

---

### Official Review · AnonReviewer1 · 2019-10-23
**Official Blind Review #1**

**Rating:** 1

**Review:**




This paper proposes domain adverarial approaches modified to address covariate shift
even in the case of shifting label distributions. The paper is interesting and the authors
are looking at an important problem, but the paper suffers several major misconceptions
and the exposition is full of errors.

The paper proposes a method called “self training” to “estimate and align” the  target label distribution
and a “prototype-based method” for conditional alignment.

The paper purports to introduce a new problem: “generalized domain adaptation”
but actually this is just covariate shift.
They confuse the term “label shift” and the colloquial “shifting label distribution”.
Moreover when they describe the method formally they fail
to even state the covariate shift assumption (that p(y|x) = q(y|x) ).
Absent this statement, the problem is unfdamentally underspecified.
They have stated only that p(y) \neq q(y) and that p(x|y) \neq q(x|y).
Thus the claimed contribution is to solve an impossible (more accurately, underspecified problem).

The proposed method leverages a “similarity classifier”
The thing that the authors call a similarity-based classifier
isn’t well explained. It looks like an ad-hoc variation on a standard
softmax prediction layer.
Here, not that similarity means similarity between x and the label weights w,
not similarity between examples and each other.
Moreover this classifier is trained as a classifier using a
standard cross-entropy objective on the target domain.
The paper lacks any formal justification for what this is offering
that we do not get from a standard classifier.

The next component of the model is "Prototype-based Conditional Alignment by Minimax Entropy”
and it is also confusingly explained.

The paper attempts to make some reference to the theory of Shai Ben David
which has been badly misapplied in the deep domain papers
(see discussion by Johannson et al 2019 and Wu et al 2019).
Strangely, the authors misattribute the theory to Zhao 2019.


There are some nice ideas in the paper and the experimental results
(flaws in domain adaptation benchmarks notwithstanding)
appear to be promising.

However the paper is written too confusingly, is outright wrong in many places
and runs the risk of badly misleading readers over even the most basic of definitions.
I encourage the authors to give the paper a gut rewrite
and do not believe that it can be published while resembling its current form.

“only aligns the covariate shift”
>>> 	Be more formal, not clear what it means to “align the shift”
	Moreover, note that owing to lack of shared support , it’s not clear
	what precisely the current methods (domain-adversarial) do
	or according to what principles they work.


“the covariate shift needs to be minimized”
>>> 	This is not a coherent way of describing the problem.
	The covariate shift is a property of the data.
	You cannot “minimize the shift”

“label shift (p(y) ̸= q(y))”
>>> 	Actually this is a “shfit in label distribution”.
	Note that you can have a shift in label distribution
	even under the covariate shift assumption.
	Label shift is the reverse assumption that p(x|y) = q(x|y).


“minimizing the label shift”
>>>	Again, this doesn’t make sense. The practitioner doesn’t get to choose the data
	that they will face at test time. The “shift” refers to the data.


"Specifically, we assume p(x|y) ̸= q(x|y) and p(y) ̸= q(y)”
>>>	The problem described in this paper is called Generalized Domain Adaptation”
	but actually it is just “covariate shift” this is not a new problem.
	The use of new terminology for old problems and misapplication of old terminology,
	e.g. “label shift” make this paper a potential danger to readers
	who then will be confused in their subsequent interactions with
	the wider literature on distribution shift.

“These methods have achieved state-of-the- art performance on several domain adaptation benchmarks“
>>>	It’s worth pointing out that benchmark SOTA is a dubious way to assess performance out of sample.
	The point is that in supervised learning you get to know that your target is the same as your source,
	so it’s ok to have a whole community smash the validation set and then see if we push the leaderboard on test data
	With domain adaption, the relevant sample size is the number of shifts, no the number of images.
	Having the community pound on 2-3 shifts tells us virtually nothing.


"our approach diminishes covariate and label shift”
>>>	Again this is not the accurate way to describe what you do.
	You attempt to salvage classifier performance under these shifts,
	you cannot “diminish the shift”.

“Recently, Azizzadenesheli et al. (2019b) propose a regularized algorithm to correct shifts in the label distribution by estimating the importance weights using labeled source data and unlabeled target data. Lipton et al. (2018b) introduce a test distribution estimator to detect and correct for label shift.”
>>>	This is not exactly the right characterization of the related work.
	The method proposed by Lipton 2018 is precisely what
	Azzizadeneshelli 2019 build upon (by adding a regularizer)

“Conventional domain adaptation approaches … only align marginal feature distribution”
>>>	Again, the authors speak about “aligning the marginal feature distribution”,
	but this is confusing, Representations are aligned, not features.


“This motivates us to align the conditional feature distribution, i.e. p(x|y) and q(x|y)”
>>>	Again these are not things that can be “aligned”. They are properties of the data.
	The entire paper needs to be re-written to be semantically coherent.

**Experience Assessment:**

I have published in this field for several years.

**Review Assessment: Checking Correctness Of Derivations And Theory:**

I carefully checked the derivations and theory.

**Review Assessment: Checking Correctness Of Experiments:**

I carefully checked the experiments.

**Review Assessment: Thoroughness In Paper Reading:**

I read the paper thoroughly.

---

> ### Author Response · Authors · 2019-11-15
> **Response to Reviewer #1**
>
> Thank you for your time and effort spent reviewing our work, and we also appreciate you pointing out the importance of the problem we are trying to solve.
>
> Also, we find your suggestions about the writing of our paper to be very helpful, thus according to them we have revised our paper and updated a new version in OpenReview.
>
> We will address your concerns below:
>
> Q1. The terms "Covariate Shift" and "Label Shift" are confusing.
> Thank you for pointing out that the terms "Covariate Shift" and "Label Shift" as used in our paper have a different meaning from the wider range of literature. In our paper, what we really mean is that we have a "shift in the feature distribution" ($p(x) \not= q(x)$) and a "shift in the label distribution" ($p(y) \not= q(y)$). To avoid confusion, we've revised these terms in our updated paper (both in the title and in content), our new title is:
> “Domain Adaptation with Feature and Label Distribution Co-Alignment”
>
> Q2. The proposed problem is just "Covariate Shift".
> Thanks for pointing out that in our paper, we did not specify an important component of our "Covariate Shift" assumption: $p(y|x) = q(y|x)$. Our proposed setting could actually be seen as a special situation of "Covariate Shift" where we also emphasize the difference between the marginal label distributions in the source and target domain. In this way, our proposed method is a more generalized version of traditional methods solving the "Covariate Shift" problem which mostly focus on the change in $p(x)$. Most methods developed for unsupervised deep domain adaptation mostly assume that $p(y)$ does not change: even though they may not state this assumption explicitly, the empirical evaluation is done on datasets that are nearly label-balanced. One of our main contributions is to show that *if* the distribution over labels does change, it can severely affect existing methods’ performance. We have updated the draft to include the $p(y|x) = q(y|x)$ assumption to clarify these points.
> The reason why we define our experiment setting as “Generalized Domain Adaptation” is based on the fact that most conventional unsupervised deep domain adaptation approaches deal with feature distributions but ignore the shift in the label distributions, which limits their generalizability to domain adaptation in real applications. This is theoretically analyzed in [1] and [2] and empirically shown by our experiments. In our setting, a model should cope *both* with the situation where only a shift of feature distribution exists *as well as* the situation where shifts in feature and label distributions occur simultaneously. In this work we try to emphasize how important it is for Domain Adaptation methods to generalize to the applications where $p(x) \not= q(x)$ while $p(y) \not= q(y)$, which are abundant in the real world. In this sense, our proposed setting is actually a more general situation than the problem that previous unsupervised deep domain adaptation methods really focused on.
>
> Q3. What is the benefit of the similarity-based classifier?
> By using the similarity-based classifier, we can encourage the representation of each sample to be closer to its corresponding weight vector, which explicitly reduces the intra-class variance in the embedding space. Therefore, we regard the weight vector for each category as the prototype of the samples of this class, which facilitates our subsequent prototype-based conditional distribution alignment.
>
> Q4. The theory of Shai Ben David is misattributed to Zhao et al. 2019.
> As Zhao clarifies (https://openreview.net/forum?id=BJexP6VKwH&noteId=HkgBK9A1sr), Eq.5 in our paper is different from the one in Ben-David et al. 2010. Furthermore, the motivation of having this theory here is to illustrate the limitation of learning domain-invariant representations, which is in line with Johannson et al. 2019 [1] and Wu et al. 2019 [2].
>
> Q5. There are some wrong descriptions in the paper.
> Thank you for pointing out this problem. We have rewritten all the related sections in the updated paper to solve this problem.
>
> Thank you again for your helpful comments. We hope that this post can address your concerns.
>
> Reference:
> [1] Support and Invertibility in Domain-Invariant Representations.  Johansson et al. AISTATS 2019.
> [2] Domain Adaptation with Asymmetrically-Relaxed Distribution Alignment. Wu et al. ICML 2019.

---

### Official Review · AnonReviewer3 · 2019-10-28
**Official Blind Review #3**

**Rating:** 1

**Review:**

In this work, the authors proposed a method to address the covariate shift and label shift problems simultaneously. In detail, the prototype-based conditional alignment and self-training based label distribution estimation is utilized. Empirically evaluation is conducted on three datasets to show the superiority of the proposed method. However, the work suffers the following weaknesses:

1). The main concern of this work is its shift assumption. In the language of dataset shift, the joint distribution p(x, y) can decompose into two different manners, which are p(y|x)p(x) and p(x|y)p(y). Covariate shift is defined as p(x) not equals to q(x), while the conditional output distribution is invariant p(x|y) = q(x|y), where p(.) and q(.) are distribution for source and target domains. Label shift is defined as p(y) not equals to q(y), while the conditional input distribution p(y|x) = q(y|x). The work assumes that p(x|y) not equals to q(x|y) meanwhile p(x) not equals to q(y). It means to minimize the joint distribution p(x, y), which is well motivated. However, it does not solve the two abovementioned shifts simultaneously here. Instead, it aims to minimize the marginal distribution and conditional distribution in the anticausal direction. See more definitions in the papers “When training and test sets are different: characterizing learning transfer” and “On causal and anticausal learning.”

2). The novelty of the paper is limited. While the authors claim that it is the first time to approach it in the proposed manner, the problem of both p(y) and p(x|y) change is not new. For instance, in the paper “Domain adaptation under the target and conditional shift,” the case of distribution shift correction also does not assume the same conditional distribution and marginal distribution for the source domain and the target domain. Also, the fulfill of conditional alignment is used the formulation and architecture of the work “Semi-supervised Domain Adaptation via Minimax Entropy,” except that there is no labeled target data in the target domain (see Eq.(1)). Besides, the notation f in Fig. 2 is missing the description of Section 3.

3).  Although the prototype-based method does help in minimizing the problem of p(x|y) not equals to q(x|y), using the minimax entropy domain adaptation in an unsupervised setting is problematic. Without a few labeled target data points, it is challenging to learn the discriminative features for the target domain. If positives and negatives (suppose it is a binary classification) are severely overlapped in the target domain,  the learned prototypes could be not consistent with those in the source domain. In other words, the prototypes might not indicate the same classes for source and target. Another issue is that the proposed model cannot solve the problem given in the assumption. In detail, the assumption is p(x, y) not equals to q(x, y), using the shared feature function F(.) and classifier C(.) for the source and the target cannot obtain domain-invariant feature and adaptive target predictor at the same time.

4). There is an issue in the label distribution by self-training. As the authors claimed, balanced sampling could diminish the effect of the label shift. However, there is no substantial theoretical evidence on this. Intuitively, the balanced sampling only ignores the original marginal distribution of the target domain. The authors should provide more explanation on it. Meanwhile, the sampling couldn’t work when there is a large number of categories. For self-training, it seems no mechanism to alleviate the label shift. Besides, the iterative learning manner heavily depends on the initialization of self-training, i.e., the top-k samples might not represent the marginal distribution.

5). For the theoretical insights, first of all, Eq.(5) is given in “A theory of learning from different domains,” Shai Ben-David et al., Machine Learning, 2010. Second, there is a mistake in Eq.(6). The second term on the right of the inequality is not JS divergence of distributions over x. Instead, it is JS divergence of those after transformation of x, z (see the subsection An information-theoretic lower bound, Zhao et al. 2019b).  Also, the descriptions of the insights are not correct.

**Experience Assessment:**

I have read many papers in this area.

**Review Assessment: Checking Correctness Of Derivations And Theory:**

I carefully checked the derivations and theory.

**Review Assessment: Checking Correctness Of Experiments:**

I assessed the sensibility of the experiments.

**Review Assessment: Thoroughness In Paper Reading:**

I read the paper thoroughly.

---

> ### Public Comment · ~Han_Zhao1 · 2019-11-06
> **Clarification**
>
> First, let me make it clear that I am not one of the authors of this paper and I don't know the authors, either. Just want to clarify a very minor point on comment 5) in the review.
>
> The citation of Eq. 5) in this paper is correct. The second term in Eq. 5) of this paper corresponds to the one using $\tilde{H}$-divergence. As a comparison, the original one in Ben-David et al. (MLJ, 2010, Theorem 1) uses the total variation (or equivalently, the $L_1$ distance). The total variation is an upper bound of the divergence defined using $\tilde{H}$ since $\tilde{H}$ corresponds to a strict subset of all the possible measurable events.
>
> On the other hand, there is indeed a slight mistake in the description of Eq. 6). $d_{\text{JS}}$ should be the Jensen-Shannon distance, not Jensen-Shannon divergence. The Jensen-Shannon distance is square root of the Jensen-Shannon divergence.

---

> > ### Author Response · Authors · 2019-11-15
> > **Thank you for your clarification!**
> >
> > We would like to sincerely thank you for your clarification, which is really helpful!
> >
> > We agree with you that the description of Eq.6 is indeed with this problem, and we have revised it in our updated paper.

---

> ### Author Response · Authors · 2019-11-15
> **Response to Reviewer #3**
>
> Q1. The shift assumption does not solve the "Covariate Shift" and "Label Shift" simultaneously.
> Same as in our response to reviewer #1, we would like to first clarify the terms in our paper. We agree that we should not use the term "Covariate Shift" and "Label Shift" in our paper, which have different meanings with the wider range of literature. In our paper, what we really meant is "the shift in feature distribution" ($p(x) \not= q(x)$) and "the shift in label distribution" ($p(y) \not= q(y)$). We've revised these terms in our updated version.
>
> Other than this, we agree with your explanation that we are trying to minimize the differences of the joint distribution in an anticasual direction.
>
> Q2. The novelty of the method is limited.
> First of all, as we mentioned in the Introduction, there indeed have been several works towards this problem [1][2][3]. While these papers provide a solid theoretical analysis of this problem under additional constraints on distributions p and q, no practical algorithm which can solve real-world cross-domain problems has been proposed ([1] and [2] are not computationally feasible for large-scale data; [3] empirically does not work well in our experiments, shown in Section 4.2). We therefore make the claim that we provided the first practical solution to this problem.
>
> Secondly, we agree the prototypical-based conditional distribution alignment used the architecture in [4]. However, we would like to emphasize that this algorithm design has a clear motivation--to align the conditional feature distribution,--and is also empirically demonstrated in Section 4 to be effective.
>
> Q3. Minimax entropy domain adaptation in the unsupervised setting is problematic.
> First of all, we agree that it is challenging to estimate the class prototypes in the target domain in an unsupervised setting. However, this is not unique to our paper and affects *all* unsupervised domain adaptation methods that aim to align the conditional feature distributions. We believe that in order to ensure the robustness of the conditional distribution alignment, additional assumptions (like the one in [3]) are needed.
> Secondly, we agree that under our proposed setting, using a shared feature function and the classifier cannot produce a single classifier that has a minimum risk on both the source and target domain,  since the source and target labels are distributed differently. To explicitly address this problem, inspired by works in label shift ([5][6]), we tried to train the classifier with weighted ERM, where importance is computed by the ratio of marginal label distributions in target and source domain ($w_i = \frac{q(y_i)}{p(y_i)}$ for each label $i$). We hoped in this way, the classifier can be optimized to better reduce the real empirical risk on the target domain. We tried to use three kinds of label distribution ratios: (1) the ratio computed by our pseudo labels; (2) the ground-truth (GT) ratio; (3) the ratio computed by the method in [5]. However, we empirically found even with the GT ratio we did not see significant benefit of this process. On the other hand, our proposed approach worked well empirically.
>
> Q4. The authors should provide more explanation on how balanced sampling and self-training could help diminish the effect of label shift.
> We agree that use of balanced sampling ignores the original distribution of the source domain but does not affect the label distribution of the target domain. However, it also helps us to learn an unbiased classifier that is used for self-training. Therefore, we tend to believe that the output results of this classifier on the target domain can better reflect the label distribution of the target domain (which is called the estimation of target label distribution in our paper). Furthermore, ideally, if we can estimate the target label distribution well, the classifier trained with the target samples with this distribution could better fit the target label distribution, as the empirical risk can better reflect the target classification risk. As discussed in the last question, we tried to explicitly use the estimated target label distribution to compute the importance weight for classifier training as in [5][6], but we did not empirically observe any benefit.
>
> Q5. The problems in theoretical insights.
> Thank you for your careful examination of our theoretical insights. Please refer to Han Zhao's clarification.
>
> Reference:
> [1] Domain adaptation under target and conditional shift. Zhang et al. ICML 2013.
> [2] Domain adaptation with conditional transferable components. Gong et al. ICML 2016.
> [3] Domain adaptation with asymmetrically-relaxed distribution alignment. Wu et al. ICML 2019.
> [4] Semi-supervised Domain Adaptation via Minimax Entropy. Saito et al. ICCV 2019.
> [5] Detecting and Correcting for Label Shift with Black Box Predictors. Azizzadenesheli et al. ICML 2018.
> [6] Regularized Learning for Domain Adaptation under Label Shifts. Lipton et al. ICLR 2019.

---

### Public Comment · ~Zirui_Wang1 · 2019-09-26
**A closely related paper**

Hi,

I found this paper to be very interesting! It might also be worth noting that a previous work has also studied the problem of negative transfer and demonstrated that matching marginal distribution alone is not sufficient to avoid negative transfer.
http://openaccess.thecvf.com/content_CVPR_2019/papers/Wang_Characterizing_and_Avoiding_Negative_Transfer_CVPR_2019_paper.pdf
Thanks.

---

> ### Author Response · Authors · 2019-09-26
> **Thank you for your helpful comment!**
>
> Hi,
>
> Thank you for your comment! [1] is a great work giving a solid analysis of negative transfer. We will cite this paper as related work.
>
> Although [1] also showed the problem of only matching marginal feature distributions, the settings of [1] and our paper are different for the following reasons:
>
> 1) Different problem settings.
> Our paper mainly focus on dealing with the domain adaptation problem when "conditional covariate shift" and "label shift" exist simultaneously, that is p(x|y)!=q(x|y) and p(y) != q(y). In contrast, [1] mainly focus on the problem of p(y|x)!=q(y|x), which was denoted as "concept shift" [2].
>
> 2) Different main ideas of solution.
> To deal with the problem that p(x|y)!=q(x|y) and p(y)!=q(y), we propose to make "domain-wise" co-alignment of conditional feature distribution and label distribution . The method in [1] on the other hand tries to use "sample-wise" gates to filter out sample x that makes p(y|x) != q(y|x).
>
> 3) Different experiment setting.
> In our paper, in order to create datasets that satisfy p(x|y)!=q(x|y) and p(y) != q(y), we propose to artificially create label shift for existed domain adaptation datasets with covariate shift. In [1], the authors used perturbation on some source samples x to make p(y|x) != q(y|x), which need to be filtered out.
>
> It is helpful to illustrate our idea by comparing our paper with [1]. Thanks for your useful suggestion!
>
> [1] Characterizing and Avoiding Negative Transfer. Zirui Wang, Zihang Dai, Barnabás Póczos, Jaime Carbonell. CVPR 2019.
> [2] An introduction to domain adaptation and transfer learning. Wouter M. Kouw, Marco Loog. Arxiv 2018.

---

> > ### Public Comment · ~Zirui_Wang1 · 2019-09-27
> > **Thanks for the author's response**
> >
> > Hi,
> >
> > I totally agree with you on the comparison especially for the problem setting. Thanks for your quick and comprehensive response!

---

### Public Comment · ~Alain_Rakotomamonjy1 · 2019-10-31
**nice work but clarifications are needed**

Thanks for this very interesting paper. I am planning to implement it but I want to be
sure to understand some points before that. please correct me if I am wrong somewhere.

I understand the part 3.1 is mostly based on the minimax entropy semi-supervised  DA of
Saito et al. so that I can use their code as a good base.

regarding the label shift contribution, it is not clear for me how self-training will help
estimate target label distribution.  I can not spot where and how you estimate
the target  label distribution in your method,  and where you use this information.

As far as I understand the paper now, it seems to me that you are just using the minimax
semi--supervised framework of Saito where labels in target domain are the pseudo-label.
I would be happy to be corrected as I am sure to have missed something.

I addition, I think it would greatly enhance the paper if you provide some ablation analysis
of the balanced sampling procedure. I think that this sampling may do most of the
job.

---

> ### Author Response · Authors · 2019-11-15
> **Thank you for your insterest in our work!**
>
> Thank you for your interest in our work!
>
> We will answer your concerns below:
>
> Q1. Where and how did you estimate the target label distribution?
> The target label distribution is estimated with our self-training process. As we have trained the classifier with class-balanced source samples, we believe the output labeling results of this classifier on the target domain can better reflect the target label distribution. Then, if we use the target samples of this label distribution to train our classifier, as the empirical risk can better reflect the target classification risk, we can alleviate the problem caused by label distribution difference.
>
> Q2. Please provide some ablation analysis of the balanced sampling procedure.
> Please refer to Table5 of our paper for this ablation study. In short, we found that balanced sampling could help all the compared methods, including ours, in most settings.

---

### Decision · Program_Chairs · 2019-12-19

**Decision:**

Reject

**Comment:**

This paper proposes a method to address the covariate shift and label shift problems simultaneously.

The paper is an interesting attempt towards an important problem. However, Reviewers and AC commonly believe that the current version is not acceptable due to several major misconceptions and misleading presentations. In particular:
- The novelty of the paper is not very significant.
- The main concern of this work is that its shift assumption is not well justified.
- The proposed method may be problematic by using the minimax entropy and self-training with resampling.
- The presentation has many errors that require a full rewrite.

Hence I recommend rejection.